# TraCo: Learning Virtual Traffic Coordinator for Cooperation with Multi-Agent Reinforcement Learning

**Weiwei Liu**
Huzhou Institute of Zhejiang University
Zhejiang University China
lww623@zju.edu.cn

**Wei Jing**
Netease Fuxi Robotics China
21wjing@gmail.com

**Lingping Gao**
Autonomous Driving Lab
Alibaba DAMO Academy China
glp.dlut@gmail.com

**Ke Guo**
Alibaba DAMO Academy
University of Hong Kong China
kguo@cs.hku.hk

**Gang Xu**
College of Control Science and Engineering
Zhejiang University China
wuuya@zju.edu.cn

**Yong Liu**
College of Control Science and Engineering
Zhejiang University China
yongliu@iipc.zju.edu.cn

**Abstract:** Multi-agent reinforcement learning (MARL) has emerged as a popular technique in diverse domains due to its ability to automate system controller design and facilitate continuous intelligence learning. For instance, traffic flow is often trained with MARL to enable intelligent simulations for autonomous driving. However, The existing MARL algorithm only characterizes the relative degree of each agent's contribution to the team, and cannot express the contribution that the team needs from the agent. Especially in the field of autonomous driving, the team changes over time, and the agent needs to act directly according to the needs of the team. To address these limitations, we propose an innovative method inspired by realistic traffic coordinators called the Traffic Coordinator Network (TraCo). Our approach leverages a combination of cross-attention and counterfactual advantage function, allowing us to extract distinctive characteristics of domain agents and accurately quantify the contribution that a team needs from an agent. Through experiments conducted on four traffic tasks, we demonstrate that our method outperforms existing approaches, yielding superior performance. Furthermore, our approach enables the emergence of rich and diverse social behaviors among vehicles within the traffic flow.

**Keywords:** autonomous driving, multi-agent reinforcement learning, counterfactual reasoning

## 1 Introduction

There are numerous Multi-Agent Systems (MAS) [1] present in nature and human society. Self-Driven Particles (SDP) [2] have been proposed to describe these systems. In SDP, each agent interacts with its surrounding agents, considers the interests of others while pursuing its own goals, and ultimately exhibits complex collective behavior. For instance, traffic flow used in autonomous driving is often considered a typical example of SDP [3]. While early SDP models were relatively simple, some have been based on philosophical ideas, such as the Belief-

7th Conference on Robot Learning (CoRL 2023), Atlanta, USA.

Desire-Intention (BDI) model [4, 5], which improves the reasoning and decision-making abilities of agents. However, traditional control methods can struggle to describe actual group behavior due to the complexity of reasoning required for artificially designed rules [6, 7].

Multi-agent reinforcement learning (MARL) [8, 9, 10] has emerged as a promising algorithm for learning controllers to simulate SDP behaviors. In the SDP of autonomous driving traffic flow, reward decomposition is particularly crucial. Each agent must strike a delicate balance between its own interests and the team's, which serves as the foundation of appropriate social behavior. However, achieving this balance is a challenging problem [11], especially in complex scenarios such as navigating busy intersections. Even experienced drivers may require the assistance of a traffic coordinator to safely navigate such environments, as

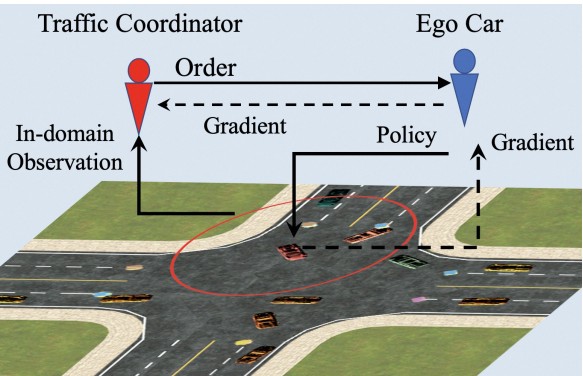

Figure 1: Example of traffic flow managed by the traffic coordinator: the vehicle operates based on its driving capabilities and follows the commands issued by the traffic coordinator.

vehicle interaction is constantly changing, making it difficult to measure each vehicle's contribution from its perspective. Thus, agents must take practical actions that prioritize the team's interests. Though previous approaches [12, 13] have attempted to evaluate an agent's contribution to the team, they often fail to address the reward balancing problem and perform poorly in traffic flow simulation applications.

We propose a novel benefit trade-off scheme inspired by the task of traffic coordination. Unlike existing schemes, we start from the needs of the team's interests and make the agent act directly according to the team's interests. As shown in Figure 1, the cars at the intersection need to follow the traffic coordinator's orders and their driving capabilities. To implement this scheme, we introduce the Traffic Coordinator (TraCo) network. TraCo learns the local interactions within a dynamic number of surrounding agents, through the Cross-Attention mechanism [14]. Therefore, TraCo is able to issue traffic order embedding based on the importance of surrounding agents relative to the feature correlation of ego agents, without altering the network structure. The agents then act according to the received orders and their states. In addition, We design the Counterfactual Advantage Function (CAF) to measure the impact of the team's orders on the team's interests, and the agent's advantage function to measure its ability to act.

The main contributions are summarized as follows:

- We propose the TraCo network to capture the interactions better and evaluate the reward decomposition of agents from the perspective of team benefits. TraCo uses a virtual traffic coordinator with a cross-attention mechanism to capture features and interactions, and issue TraCo commands within the traffic scene, thereby assisting agents in generating strategies to act according to the interests of the team.

- We incorporate CAF in TraCo to measure the impact of TraCo's commands to the agent on the team's interests. This allows agents to act directly in line with the team's interests, while also promoting TraCo's ability to move beyond the limitations of a feature extraction network, and thus significantly improve the performance of the simulated traffic flow.

- We conduct extensive experiments. The results show that TraCo performs superior in multiple metrics, and the agents exhibit diverse social behaviors.

# 2 Related Work

## 2.1 MARL and Value Decomposition

In a multi-agent system, the agents share the same reward function. As a result, the rewards they obtain may not accurately reflect their behaviors, leading to inaccurate policy updates. This is known as the credit assignment problem [11] in MARL.

In discrete action spaces, value decomposition is a solution. VDN [15] uses a simple summation to calculate the joint action-value function for decomposing the team's reward signal to each agent. Compared with IQL[16], this centralized training method can guarantee the optimality of the joint action value function to a certain extent. Nevertheless, the simple summation significantly limits the fitting ability of the joint action-value function. QMIX [13] improves the fitting process of simple summation in VDN to non-linear fitting subject to monotonic constraints. QTRAN [17] introduces an intermediate action-value function that approximates the real action-value function and then decomposes the intermediate action-value function, which avoids the monotonic constraint of QMIX. In addition, given the limitation of the QMIX action-value function fitting ability with monotonic constraints, the agent cannot explore the entire joint action space. MAVEN [18] shares a hidden variable in the value function of each agent, and uses the mutual information obtained by maximizing the trajectory information of the agent and the hidden variable information to increase the divergence of the policy and make the actions more diverse.

For continuous action state spaces, existing work often learns the joint state-value function directly. One such algorithm that does this is MADDPG [19], which extends the DDPG [20] algorithm to the multi-agent system. MADDPG adopts a scheme of centralized training and decentralized execution, where the joint value function is computed and then used to evaluate the policy of each agent. Another algorithm that follows a similar path is MAPPO [21], which extends PPO [22] to the multi-agent domain. Interestingly, MAPPO requires only a minimal hyperparameter search to achieve comparable performance to state-of-the-art algorithms. Both MADDPG and MAPPO implicitly solve the multi-agent credit assignment problem. In contrast, COMA [23] tackles the credit assignment problem head-on by using a counterfactual baseline to evaluate the contribution of each agent to the team. In addition, simple independent learning[24] that solely pursues self-interest may make traffic vehicles aggressive and display irrationally selfish behavior. Unlike existing algorithms that measure their contribution to the team starting from the agent, distributing the team's needs directly to the agents could be an interesting solution.

## 2.2 Autonomous Traffic Flow

Collecting data solely from the real world for autonomous driving is impractical and expensive. Therefore, traffic flow simulations have emerged as a popular alternative for modeling vehicle interactions. Traditional approaches [25] employed predetermined rules to control traffic flow. Recent research has introduced RL as a way to control vehicles in traffic flow. For example, CityFlow [26] employs RL algorithms to study traffic flow on a city-wide scale. RL has also been used to train individual vehicles [27] in controlled environments and examine vehicle-to-vehicle social interactions [28]. Additionally, SMARTS [29] studies the interaction capabilities between agents in different environments. Nevertheless, the study of interactions between agents in diverse environments remains an area of interest. In this context, our research investigates four common autonomous driving scenarios, with a focus on applying the traffic coordinator network in continuous action spaces. Our approach starts with teams, which directly ask agents to act in favor of the team, explicitly addressing the trade-off between team and self-interest, which allows us to model the traffic flow more efficiently than previous work.

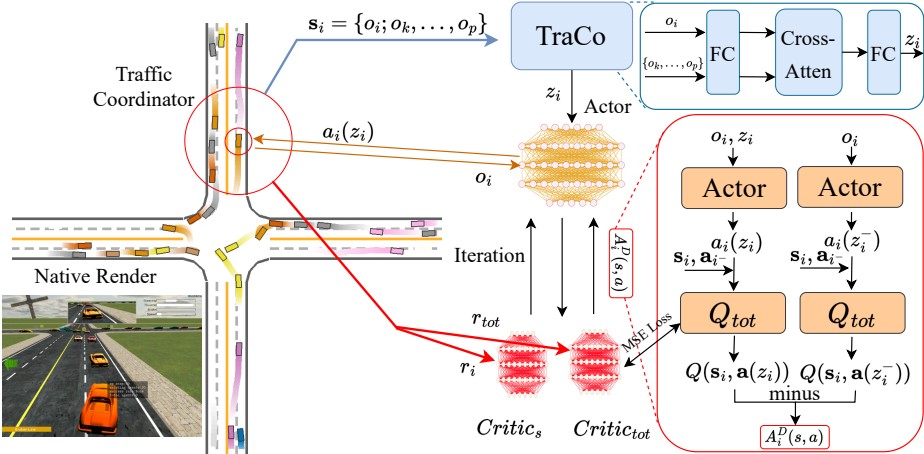

Figure 2: The architecture of the Traffic Coordinator network. The left image showcases the simulated environment, with the third-person perspective visible in the bottom left corner. The observations of agents around agent $i$ are denoted as $\{o_k, \cdots, o_p\}$, while $o_i$ represents the observation of agent $i$. We incorporate two evaluation networks ($Critic_s$ and $Critic_{tot}$) to fit individual and joint state value functions, respectively. Equations (10) and (11) illustrate how the joint action-value function can be computed via the joint state-value function, followed by using MSE loss to train the action-value network.

## 3 Methods

Autonomous vehicles must be vigilant to avoid collisions while reaching their destination. In reality, drivers drive carefully under the guidance of the traffic coordinator, which provides clear driving directions based on the surrounding situation, reducing the driving difficulty. This section provides a detailed overview of the TraCo network, which enhances the social behavior of traffic flow. To ensure that compliance with TraCo's orders improves the team's interest, we utilize the counterfactual advantage function. The TraCo network differs from traditional centralized algorithms by reducing the information processing difficulties on each agent and providing clear centralized vehicle guidance.

### 3.1 Traffic Coordinator

As shown in Figure 2, We introduce a virtual agent, the traffic coordinator, to model the traffic flow. The traffic coordinator has in-domain observations and distributes order vectors $z^i \in \mathbb{R}^{d_z}$ ($z^i$ is the order issued by the traffic coordinator to agent $i$, and $d_z$ is the order dimension) to each agent. The orders issued by the traffic coordinator network can be represented by $\mathbf{z} = z^i \mid i \in N$, where $N$ is the number of agents. The function $f$ generates the order $\mathbf{z}$ and is parameterized by $\varphi$, with $z_i \sim f_\varphi(o_i, \mathbf{s}_i)$, where $o_i$ is the observation of agent $i$, and $\mathbf{s}_i$ is obtained by aggregating agent information in the domain, as illustrated in Figure 2. Upon receiving the traffic coordinator order $z_i$, each agent $i$ takes actions according to its observation $o_i$. In an episode, the traffic coordinator observes the state in the domain, calculates, and distributes orders $\mathbf{z}_t$ to the agent. At each moment, any agent $i$ will act based on its individual observation and order $z_i$, $p_i = \pi(o_i, z_i)$, $a_i \sim p_i$. The objective function is formulated using PPO as follows:

$$\mathcal{L}(\theta) = \Sigma_i^n \min(\frac{\pi_\theta(a_i \mid o_i, z_i)}{\pi_{\theta'}(a_i \mid o_i, z_i)} A_i, cilp(\frac{\pi_\theta(a_i \mid o_i, z_i)}{\pi_{\theta'}(a_i \mid o_i, z_i)}, 1 - \varepsilon, 1 + \varepsilon)A_i), \quad (1)$$

Since the information closer to the vehicle may be more important, as shown in Figure 2, In order to capture the interactions between agents, we utilize a Cross-Attention network [14] in the traffic coordinator network to extract relevant information.

## 3.2 Counterfactual Advantage Function

Updating the policy solely based on Equation (1) results in the traffic coordinator network issuing orders only to extract in-domain features, with the agent action network making more comprehensive decisions based on these features. The advantage function remains unchanged as follows:

$$A(o_i, z_i, a_i) = Q(o_i, z_i, a_i) - V(o_i, z_i)$$
$$= E_{s_{t+1} \sim p(s_{t+1}|s_t, a_t)}[r(s_t) + \gamma V^{\pi}(s_{t+1}) - V^{\pi}(s_t)] \tag{2}$$

However, this approach fails to address the critical interests balance problem. Since traditional TD errors only consider agent rewards, the contribution of each agent to the team is still not measured. Therefore, inspired by the difference rewards[30], we design counterfactual rewards to measure the impact of whether each agent acts according to the team's order on the team's interests.

$$r_i^D = r_{tot}(o_i, a_i(z_i), \mathbf{o}_{i-}, \mathbf{a}_{i-}) - r_{tot}(o_i, a_i(z_i^-), \mathbf{o}_{i-}, \mathbf{a}_{i-}), \tag{3}$$

Here $\mathbf{o}_{i-}$ and $\mathbf{a}_{i-}$ represent the joint observations and joint actions of agents other than agent $i$, $a_i(z_i^-)$ represents the action of the agent only based on its own observation when there is no traffic coordinator order. $r_{tot}$ represents the team reward in the domain, $r_{tot} = r_k^s + \cdots + r_p^s, \{k \cdots p \in domain_i\}$. At this point, the reward for agent $i$ is:

$$r_i = r_i^s + r_i^D, \tag{4}$$

At this point, agent $i$'s reward comes from its reward, and the team's reward. According to Equation (3), obtaining $r_i^D$ needs to perform actions with or without traffic coordinator orders so that the environment gives different reward signals, which requires modeling the environment. Obviously, this process is overly complicated. Researchers [31] suggest using function approximations instead of simulators to estimate differential rewards.

Similar to COMA [23], we design a centralized critic, which is used to estimate the joint actions value function $Q_{tot}$ of all agents in the domain. Then, for each agent $i$, with this value function $Q_{tot}$, we can compute a Counterfactual Advantage Function (CAF), keeping the actions $\mathbf{a}_{i-}$ of other agents fixed in the process:

$$A_i^D(\mathbf{s}, \mathbf{a}) = Q_{tot}(\mathbf{s}_i, \mathbf{a}(z_i)) - Q_{tot}(\mathbf{s}_i, \mathbf{a}(z_i^-)) \tag{5}$$

In Equation (5), a centralized critic is employed to reason about counterfactuals. Specifically, the critic considers the scenario where only the actions of agent $i$ change and computes the counterfactual advantage function $A_i^D(\mathbf{s}_i, \mathbf{a}_i)$, which reflects the contribution of agent $i$ to the team. This approach enables learning directly from the agent's experience without relying on an additional environment model.

Based on this, the advantage function of agent $i$ at this time is:

$$A_i = A_i^s + \alpha * A_i^D \tag{6}$$

$$A_i^s = Q_i(o_i, a_i) - V_i(o_i)$$
$$= \mathbb{E}[r_i^s + \gamma V_i^{\pi} - V_i^{\pi}] \tag{7}$$

where $\alpha$ represents the coefficient of the counterfactual advantage function. $A_i^s$ is the advantage function obtained by agent $i$ based on its reward function. Please see the appendix for additional methods and pseudocode.

# 4 Experiments

## 4.1 Baseline Algorithms and Experiment Setup

Experimental baseline algorithms include IPPO [24], MFPO [32], and CoPO [3]. IPPO uses PPO as an independent learner; MFPO encodes the state of surrounding agents as a mean state, which

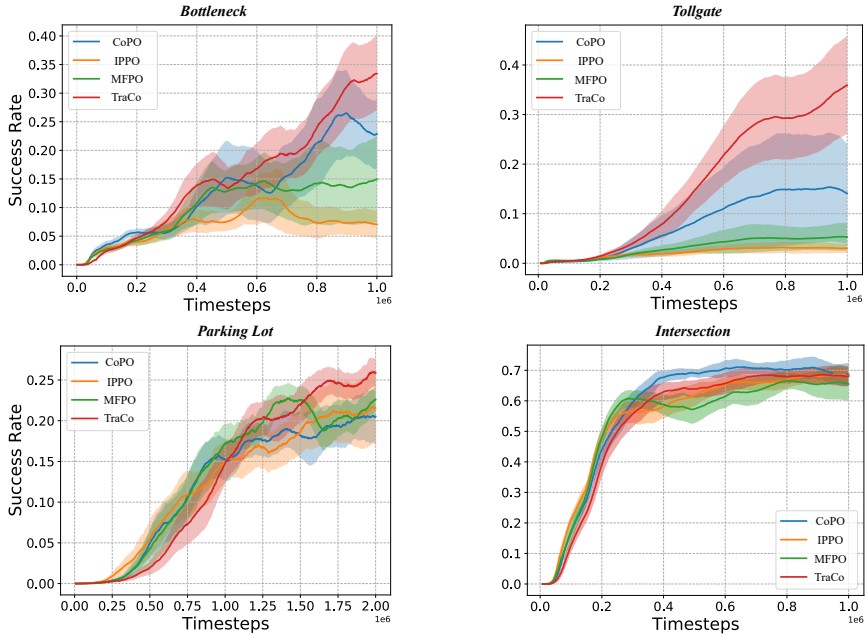

Figure 3: Performance comparison of TraCo and other baseline algorithms in four task scenarios. Note that the time interval step of the features extracted by the traffic coordinator network is 1.

is used as an auxiliary input to the value function. CoPO [1] is to split the source of the agent into its reward and the average reward of the surrounding agents, and the global reward controls the proportion of the two.

We ran experiments using RLlib [33] with the aforementioned environments and algorithms on 4 Nvidia GeForce RTX 2080Ti GPUs. Each trial was trained on over 1 million environment steps, equivalent to approximately 55 hours in a real-time traffic system or over 2,000 hours of individual driving experience, assuming an average of 40 vehicles running at the same time. For the introduction of the experimental scenarios and tasks, please refer to the appendix.

## 4.2 Results

In Figure 3, we compare the success rates of our TraCo algorithm to those of the baseline algorithms across all tasks. As a result of the virtual traffic coordinator's order, our TraCo algorithm outperforms the baseline in three tasks and performs comparably to the baseline only in the Intersection task. Notably, in the Tollgate task, which involves the most agents, TraCo outperforms the strong baseline CoPO by a significant margin. As shown in Figure 5, this task requires agents to exhibit active cooperation behaviors, such as queuing and giving way, and strong interaction abilities among agents. Populations generated by other algorithms fail to exhibit such behavior, leading to congestion. Interestingly, IPPO performs comparably to MFPO and even outperforms CoPO in the Parking Lot task, despite MFPO and CoPO having more intra-domain information. This is because the value estimated by IPPO's critic network includes noise perturbations that improve the algorithm's exploration performance. Additionally, the Parking Lot task is continually changing due to community factors. Simply averaging or concatenating neighbors' states as an additional input to the value function makes training unstable, a phenomenon also observed in MARL from StarCraft [24, 34].

---

[1]https://github.com/metadriverse/metadrive-benchmark/tree/main/MARL; As described in the repo link, CoPO measures performance using the maximum value of each set of experimental data with different random seed. However, for more comprehensive comparison, we use the average performance across 8 random seeds in this study.

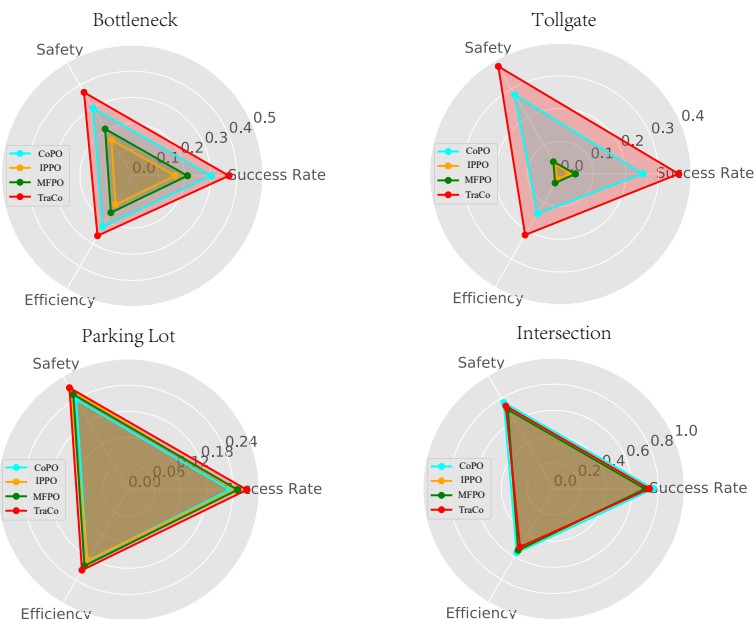

Figure 4: Performance comparison among TraCo and baseline algorithms with three metrics.

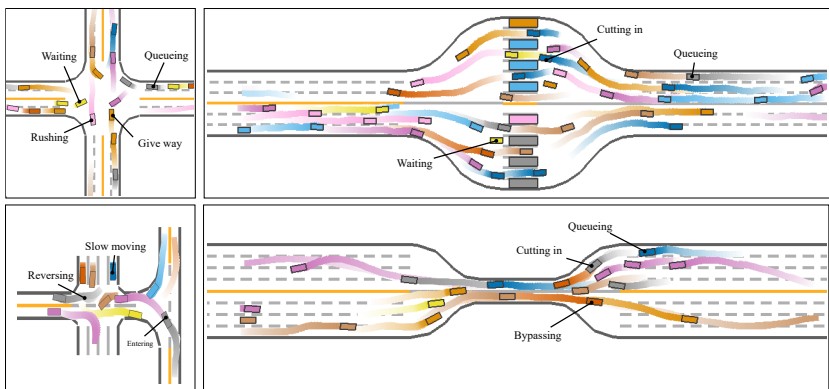

Figure 5: Visualization of the social behavior of the population with TraCo. The social behavior of each agent is denoted with black dots, while the subsequent trajectory of each vehicle is indicated by a decreasing intensity of color, with brighter colors representing more recent steps.

Figure 4 depicts the lidar chart with three metrics after normalization. Despite the comparable success rate, TraCo outperforms baseline algorithms on safety and efficiency metrics. To render the vehicle running track, MateDrive employs PyGame. Figure 5 illustrates that TraCo generates populations that exhibit social behaviors, such as reversing, cutting in line, queuing, waiting, and following, in all four tasks to complete their goals. This demonstrates that the vehicle selects different driving styles based on the situation and simulates a range of interactive behaviors in the traffic system.

## 4.3  Generalization

To evaluate the generalization ability, we vary the initial number of agents in the test phase to determine their converged policies. Figure 6 illustrates that as the number of agents increases, the population success rate decreases due to road congestion and a higher likelihood of collisions. However, we observe that having too few agents did not improve the algorithm's performance in the Intersection task. We suspect that in the multi-agent algorithm, each agent's policy may overfit the behavior

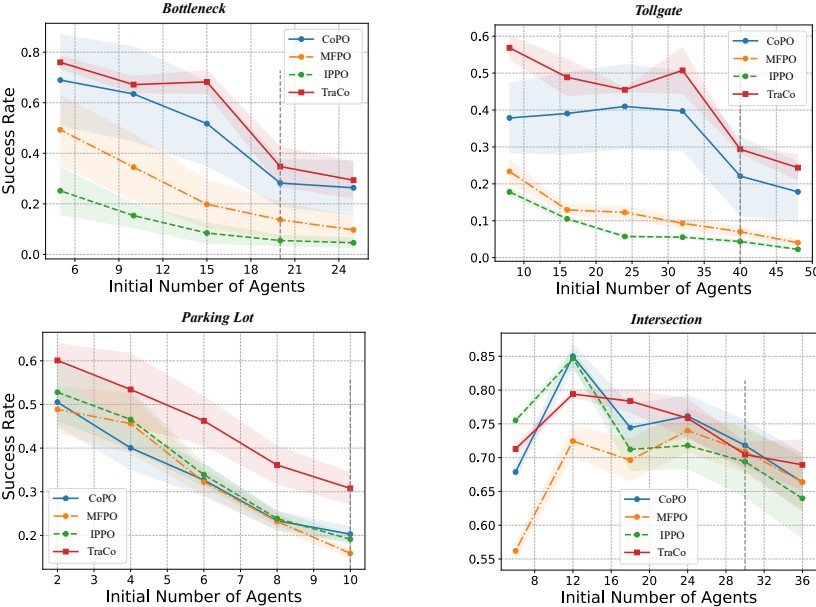

Figure 6: Success rate for different initial numbers of vehicles at test time. The gray vertical line represents the initial number of vehicles during training, from which the algorithm policy is trained.

of other agents, resulting in failure. This overfitting may occur because a reduced number of agents leads to fewer encountered situations, which can limit the model's ability to generalize. Additionally, we find that inputting the reward distribution coefficient, which is learned during CoPO training, as prior knowledge into the agent observation may interfere with the generalization ability of the algorithm once the number of agents changes during the test phase. Notably, TraCo outperforms baseline algorithms, even when the number of agents in the population changes. This is because TraCo uses a cross-attention network to process the dynamic number of agent information in the domain, allowing its model to adapt to the community environment of the dynamic number of agents. Finally, we have designed ablation experiments, which are detailed in the Appendix.

# 5 Limitation

TraCo has demonstrated its ability in traffic flow simulation by facilitating team instructions, while agents exhibit complex social behaviors. However, there are still gaps in its ability to replicate real-world vehicle behaviors, which can be attributed to the random exploration of reinforcement learning. To address this shortcoming, we plan to integrate real data into our approach in the future, in order to constrain vehicle behavior and improve overall performance. In addition, TraCo still lacks the ability to handle traffic lights, which may limit its application to certain urban driving scenarios.

# 6 Conclusions

We present a novel approach to model traffic flow using the Traffic Coordinator Network (TraCo) with the Counterfactual Advantage Function (CAF) and an attention mechanism. TraCo models real traffic coordination to enhance vehicle decision-making unlike traditional feature extraction networks. Our experiments demonstrate that TraCo-trained vehicles exhibit lower collision rates and higher success rates than baseline models while demonstrating a diverse range of social behaviors.

# 7 Acknowledgment

This work is supported by NSFC 62088101 Autonomous Intelligent Unmanned Systems.

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

# 8 Appendix

## 8.1 Method Supplement

In order to calculate the counterfactual advantage function $A_i^D$, the joint action value function $Q_{tot}$ must be estimated, although only the state value function $V_{tot}$ is estimated with PPO. Thus, we have:

$$V_{tot}^{\pi,\gamma}(s_t) := \mathbb{E}_{s_{t+1}:\infty;a_t:\infty}\left[\sum_{l=0}^{\infty}\gamma^l r_{t+1}^{tot}\right], \tag{8}$$

and:

$$Q_{tot}^{\pi,\gamma}(s_t, a_t) := \mathbb{E}_{s_{t+1}:\infty;a_{t+1}:\infty}\left[\sum_{l=0}^{\infty}\gamma^l r_{t+1}^{tot}\right]. \tag{9}$$

From equations (8, 9), we get:

$$\begin{aligned} Q_{tot}^{\pi,\gamma}(s_t, a_t) &:= r_t^{tot} + \gamma\mathbb{E}_{s_{t+2}:\infty;a_{t+1}:\infty}\left[\sum_{l=0}^{\infty}\gamma^l r_{t+2}^{tot}\right] \\ &= r_t^{tot} + \gamma V_{tot}^{\pi,\gamma}(s_{t+1}) \end{aligned} \tag{10}$$

In this way, we can obtain the action-value function $Q_{tot}$ from the state-value function $V_{tot}$ in the PPO architecture. However, since state-value functions cannot evaluate agent actions, we design an action-value function network that takes input approximating Equation (10). To train the network, we utilize Mean Square Error (MSE) loss [35]:

$$\begin{aligned} MSE &= \frac{1}{b}\sum_{i=1}^{b}(y_i - y_i')^2 \\ &= \frac{1}{b}\sum_{i=1}^{b}\left(Q_{tot}^{net} - r_t^{tot} - \gamma V_{tot}^{\pi,\gamma}(s_{t+1})\right)^2. \end{aligned} \tag{11}$$

where $b$ denotes the batch size. Algorithm 1 shows the overall process of TraCO in the appendix.

At this time, the partial differential of the counterfactual advantage function for agent $i$ to take an action under instruction $z_i$ is:

$$\begin{aligned} \frac{\partial}{\partial z_i}A_i^D &= \frac{\partial}{\partial z_i}Q(\mathbf{s}_i, \mathbf{a}(z_i)) - \frac{\partial}{\partial z_i}Q(\mathbf{s}_i, \mathbf{a}(z_i^-)) \\ &= \frac{\partial}{\partial z_i}Q(\mathbf{s}_i, \mathbf{a}(z_i)) \end{aligned} \tag{12}$$

The above equation reveals that the agent $i$'s utility with counterfactual instructions aligns with the global learning objective, and maximizing the counterfactual reward can enhance the joint action-value function. Consequently, the agent's advantage function can be decomposed into its individual and team contributions, as shown in Equation (6), which completes the value decomposition operation.

**Algorithm 1** TraCo for agent $i$

---

1: **Input:** Randomly initialize TraCo, actor and critic network $f$, $\pi$ and $V$ with weights $\varphi$, $\theta_\pi$ and $\theta_v$
2: **for** episode=1, $T$ **do**
3:      Get agents' observations $\{o_1, \cdots, o_n\}$
4:      Get $s_i = \{o_k, \cdots, o_p\}$ according to the distance
5:      Compute $z_i = f(o_i, s_i)$, $a_i = \pi(o_i, z_i)$
6:      Compute counterfactual advantage function $A_i^D$ according to equations (5, 10, 11)
7:      Compute $A_i$ according to equation (6)
8:      Update with PPO rules
9: **end for**

---

## 8.2 Experiment Platform and Scenarios

We use MetaDrive [36] as a simulator, which is capable for generating infinite scenarios with various road maps and traffic settings to enable generalizable RL. In our setup, we use current state, navigation info, and surrounding data encoded in a vector of 72 lidar-like measurements as agent observations, while the policy output is the acceleration and steering of the vehicle. As the mutual influence between vehicles decreases with distance, we define the in-domain state for the traffic coordinator as the information splicing of different vehicles within a 40-meter radius of the ego-vehicle.

As shown in Figure 5, we benchmark our method in four common autonomous driving tasks, which are described in detail as follows:

**Bottleneck**: The Bottleneck is to set up a narrow bottleneck lane between the eight lanes, forcing vehicles to give way and queue up to pass. The environment is initialized with 20 cars.

**Tollgate**: The Tollgate environment models the real-world behavior of vehicles passing through a tollgate, where agents are required to wait for a permission signal for 3 seconds before continuing. Failure to comply with this rule results in a failed episode. The environment is initialized with 40 cars.

**Parking lot**: The parking lot scenario in our simulation consists of 8 parking spaces. Spawn points for vehicles are scattered both within and outside the parking lot, leading to simultaneous entry and exit of vehicles and thereby increasing the level of difficulty. The environment is initialized with 10 cars.

**Intersection**: At an unprotected intersection scenario, vehicles are required to negotiate and judge the potential intentions of other parties in order to complete the task. The environment is initialized with 30 cars.

In this paper, we use three indicators to evaluate the performance of multi-agent algorithms. *success rate* is the ratio of vehicles successfully reaching the destination, *safety* is the vehicle non-collision rate, *efficiency* $>= 0$ indicates the difference between successes and failures in a unit of time $(N_{success} - N_{failure})/T$. Vehicles may travel at low speeds for the safety of driving, but this is not conducive to the effective passage of vehicles.

## 8.3 Ablation Studies

In our previous experiments, we employed the traffic coordinator network solely as a feature extraction network, without considering the counterfactual advantage function. Therefore, it is crucial to verify the validity of this function. As illustrated in Figure 7, TraCo w/o CAF performs worse than TraCo w/ CAF in all four autonomous driving tasks. This is because the traffic coordinator network, when equipped with a counterfactual advantage function, not only extracts in-domain features but also evaluates the agent's behavior based on these features. This evaluation allows for the

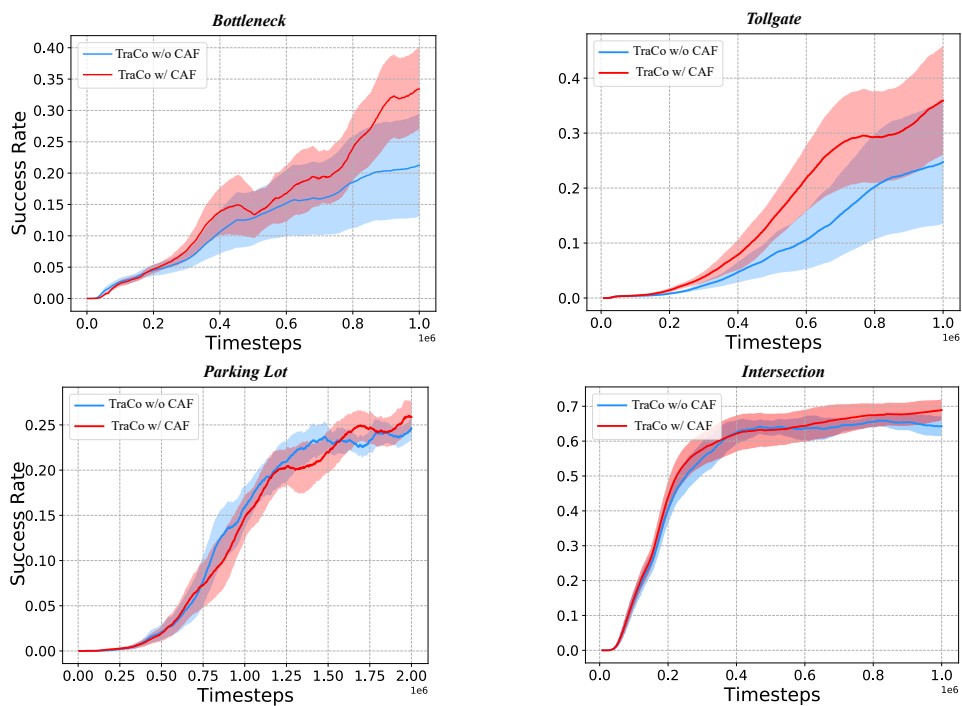

Figure 7: Performance comparison of TraCo with and without counterfactual advantage functions.

Table 1: The traffic coordinator network re-issues the command $z$ according to the current situation at different time intervals, and the command $z$ remains unchanged during this time interval.

| | Bottleneck | | | Tollgate | | | Parking lot | | | Intersection | | |
|---|---|---|---|---|---|---|---|---|---|---|---|---|
| | Success Rate | Efficiency | Safety | Success Rate | Efficiency | Safety | Success Rate | Efficiency | Safety | Success Rate | Efficiency | Safety |
| TraCo/1 | 0.36 ±0.13 | 0.26 | 0.36 | **0.36 ± 0.19** | **0.22** | **0.38** | **0.27 ± 0.04** | **0.21** | **0.27** | 0.73 ± 0.05 | 0.51 | 0.73 |
| TraCo/2 | 0.37 ± 0.09 | 0.27 | 0.37 | 0.32 ± 0.15 | 0.18 | 0.34 | 0.15 ± 0.07 | 0.11 | 0.16 | 0.72 ± 0.01 | 0.51 | 0.72 |
| TraCo/4 | 0.38 ± 0.07 | 0.28 | 0.38 | 0.17 ± 0.16 | 0.09 | 0.2 | 0.14 ± 0.06 | 0.11 | 0.15 | 0.72 ± 0.03 | 0.51 | 0.72 |
| TraCo/6 | **0.42 ± 0.09** | **0.3** | **0.42** | 0.25 ± 0.19 | 0.14 | 0.28 | 0.17 ± 0.04 | 0.13 | 0.17 | 0.73 ± 0.04 | 0.51 | 0.73 |
| TraCo/8 | 0.34 ± 0.12 | 0.25 | 0.34 | 0.29 ± 0.18 | 0.16 | 0.31 | 0.21 ± 0.04 | 0.16 | 0.21 | **0.74 ± 0.02** | **0.53** | **0.74** |

measurement of the agent's contribution to itself and the surrounding team, effectively addressing the interests balance problem.

Taking inspiration from the behavior of real-life traffic coordinators, who issue commands based on vehicle behavior and intersection information at time intervals rather than continuously directing vehicles, we designed different time intervals for the Traffic Coordinator Network (TroCo) to extract features. As shown in Table 1, our experiments reveal that in complex traffic environments such as Tollgate and Parking lot, where obstacles are numerous, roads are congested, and the behavior of domain agents is difficult to predict, frequent direction is necessary to ensure optimal vehicle decision-making. However, in Bottleneck and Intersection tasks, where the purpose of the vehicle is clear, and the behavior is more predictable, frequent direction may interfere with the agent's decision-making. In such cases, an appropriate time interval can enhance the consistency of the agent's behavior.

