# OpenReview forum: "TraCo: Learning Virtual Traffic Coordinator for Cooperation with Multi-Agent Reinforcement Learning"
_robot-learning.org/CoRL/2023/Conference — CoRL 2023 Poster_

### Official Review · Reviewer_9NNb · 2023-07-12

**Confidence:** 4
**Originality:** Good
**Technical Quality:** Fair
**Clarity Of Presentation:** Fair
**Impact:** 3

**Recommendation:**

Weak Reject: I recommend rejecting the paper, but will not argue for my recommendation if the majority of other reviewers have a different opinion.

**Review:**

There are many details that need to be explained in detail by the author.

1. What are the functional differences between the coordinator and the traffic light? Why not train the signal light directly？

2. Overall, the method description is quite confusing. Please provide a detailed description of the method and explain why it can solve the problem of "cannot express the contribution that the team needs from the agent.

3. The method is similar to HRL, where the top layer provides goal z, while the bottom layer makes decisions based on z from individual rewards and group rewards. The author needs to introduce the top-level part in detail, i.e., the blue section of TraCo in Figure 2. The author provided a relatively detailed definition of Ai, but from the structure in Figure 2, it is not the main part of TraCo.

4. As it is an application-oriented problem, the author also needs to introduce the model definition of the environment in traffic scenarios, including observation, actions, and two rewards.

5. Are the two actors and two Qtots in Figure 2 the same? How to estimate Q(s, a(z-))? Where does its data come from? How to obtain r and s' corresponding to a(z-)?

6. The definitions of Qtot and Qnet in the article need to be introduced. Different symbols are used for the same definition before and after the article.

7. How to set the hyperparameter alpha in formula 6

8. Why is Figure 4 only showing three scenarios without another one?

9. Need ablation experiment to demonstrate the effectiveness of the proposed counterfactual strategy in training the Q(s,a(z))-Q(s,a)

**Quality Of The Limitations Section:**

Additional details required

**Questions For Rebuttal:**

Please refer to the pros and cons part.

**Robotics Focus:**

Highly relevant to robotics but no hardware experiments

**Summary Of Paper:**

This paper proposes TraCo, which utilizes a combination of cross-attention and counterfactual advantage function to learn a virtual traffic coordinator for cooperation with multi-agent reinforcement learning. The goal is to accurately quantify the contribution that a team needs from an agent in the field of autonomous driving and to enable the emergence of rich and diverse social behaviors among vehicles within the traffic flow. There are many details that need to be explained in detail by the author.


**Summary Of Recommendation:**

There are many details that need to be explained in detail, thus leaning to a rejection.

---

### Official Review · Reviewer_nkFo · 2023-07-19

**Confidence:** 3
**Originality:** Good
**Technical Quality:** Very Good
**Clarity Of Presentation:** Good
**Impact:** 4

**Recommendation:**

Weak Accept: I recommend accepting the paper, but will not argue for my recommendation if the majority of other reviewers have a different opinion.

**Review:**

Originality) The paper introduces a novel approach to modeling traffic flow by utilizing the Counterfactual Advantage Function and cross-attention mechanism. This approach surpasses existing solutions and promotes social behavior, effectively reducing collision rates.

Quality) The submission is technically convincing. The experiment results are well-described and persuasive.

Clarity ) The paper does not provide enough information about problem formulation, action and observation space as well as algorithm and hyperparameters, however authors state that there will be more details about it the in appendix. I found a few typos, which will be listed in the questions section

Significance) The paper and its results are important and can be used by other researchers to further improve traffic flow simulation. The paper improves the safety and success rate of vehicle agents.

Relevance) I think it is a relevant paper because it outperforms existing solutions in traffic flow simulation. This paper brings us closer to a future where such experiments can be conducted in real-world scenarios.

Limitations) I think limitations are addressed well


**Quality Of The Limitations Section:**

Limitations are addressed clearly

**Questions For Rebuttal:**

•	Close round-bracket in Equation (1)

•	At line 204 – “fro”

•	I miss at least a brief description of observation and action space as well as problem formulation and experiment description

•	I would suggest moving equations in the appendix and addressing missing parts

**Robotics Focus:**

Relevant but unlikely to deploy to hardware in near future

**Summary Of Paper:**

The authors consider the problem of managing traffic flow and propose an algorithm with a virtual agent that serves as a traffic coordinator, sending orders to other individual agents. The authors introduce the Counterfactual Advantage Function (CAF) to motivate agents to work in line with the team's interests. Additionally, they utilize cross-attention to effectively handle dynamic numbers of observations and agents. In their experiments, they conduct experiments using RLLib and demonstrate that their algorithm outperforms existing approaches in four different traffic flow simulations

**Summary Of Recommendation:**

It is an important paper that demonstrates the successful approach to modeling traffic flow and showcases the emergence of social behavior among vehicles. It is written clearly but certain sections that are crucial for understanding are missing. Additionally, I noticed a few typos and proposed to write more information about problem setting and experiments. After all, I find this paper significant and relevant to the conference

After author response: I appreciate the author's rebbutal. However I miss the final draft to change my score.

---

### Official Review · Reviewer_qaUa · 2023-07-20

**Confidence:** 5
**Originality:** Excellent
**Technical Quality:** Excellent
**Clarity Of Presentation:** Excellent
**Impact:** 4

**Recommendation:**

Strong Accept: I recommend accepting the paper and will argue for my recommendation even if other reviewers hold a different opinion.

**Review:**

This paper is a prime example of high-quality academic writing. All figures are informative and plotted clearly. Reading this paper is an intellectual pleasure. I believe this is a high-quality paper in clarity, originality, and significance. And I don’t have many suggestions.

Strengths:
1. The logical flow of this paper is well-executed, facilitating a comprehensive understanding of the presented ideas.

Suggestions:
1. The authors could mention some details in the TraCo model training process, including the training environment setting.

**Quality Of The Limitations Section:**

Limitations are addressed clearly

**Questions For Rebuttal:**

Questions:
1. The authors could elaborate on the agents' observations, states, and actions under different experiment settings. What are the observations, states, and actions for the four experiment scenarios?
2. The authors mentioned three metrics in Section 4.2 and Fig 4. However, the authors should have explained how the other two metrics, safety and efficiency, are calculated.
3. While the authors cited the original paper, the authors could explain the details of the cross-attention mechanism.
4. Can the model handle external impacts on traffic patterns? And what could be the corresponding changes in MARL agent configurations?

**Robotics Focus:**

Highly relevant to robotics but no hardware experiments

**Summary Of Paper:**

This paper introduces the Traffic Coordinator Network (TraCo), a novel method for addressing Multi-agent Reinforcement Learning (MARL) limitations and the context of autonomous driving. Traditional MARL systems fail to accurately evaluate an agent's contribution to the team, which may change over time. The TraCo approach in this study resolves this issue by focusing on team needs and interests. TraCo uses a cross-attention mechanism and a Counterfactual Advantage Function (CAF) to understand distinctive characteristics of domain agents and accurately quantify an agent's contribution. This approach facilitates improved decision-making in traffic flow simulations. While there are still gaps in replicating real-world vehicle behaviors due to reinforcement learning's random exploration, this approach significantly outperforms existing models in various metrics.

**Summary Of Recommendation:**

This paper presents the Traffic Coordinator Network, a pioneering solution to address the shortcomings of Multi-agent Reinforcement Learning in autonomous driving. This is a high-quality paper, and I recommend a strong acceptance.

---

### Official Review · Reviewer_TCct · 2023-07-23

**Confidence:** 3
**Originality:** Good
**Technical Quality:** Fair
**Clarity Of Presentation:** Fair
**Impact:** 2

**Recommendation:**

Weak Reject: I recommend rejecting the paper, but will not argue for my recommendation if the majority of other reviewers have a different opinion.

**Review:**

This paper seems to be about how to control a large number of vehicles on a busy road.
It is interesting to note that the methodology approach in this study is based on multimodal reinforcement learning, where the overall reward can be decomposed as the reward of individual vehicles.
However, the environmental settings used in the experiments are simple (such as a nondescript crossroads or a bottlenecked power line road) and may not be sufficiently validated to be effective in a wide variety of traffic situations.


**Quality Of The Limitations Section:**

Limitations are addressed clearly

**Questions For Rebuttal:**

The critical factors in adapting multi-agent reinforcement learning to self-driving vehicles appear to be learning speed and safety.
A large data set would be needed for learning. It is unclear how the proposed approach would work for this problem.
Could it sustain better performance with less data than other existing methods?

In addition, there are questions about whether multi-agent reinforcement learning is really appropriate for solving the challenges of automated driving in traffic situations like this one.
The proposed method has the advantage of being able to split the reward into individual vehicles.
However, some disadvantages may also exist.
For example, there is the issue of quantity and variation of the data set.
In addition, the proposed method seems to only compare with other reinforcement learning methods, and no comparison has been made with methods that control the entire system centrally or with other heuristic methods.
Wouldn't it be imperative for the authors to compare the proposed method with these other approaches in order to evaluate it from a broader perspective?

I have some questions about the experiment. As supplementary material, the authors have attached a video. In this video, the vehicles were seen crashing. Afterward, the colliding vehicle completely disappeared.
This is very different from the real case. Does the disappearance of the car all at once after the collision cause any problems in learning the model?
It is also unclear how the data that had been acquired from the vanished car is handled.

Additionally, cars were seen crossing the white line. In automated driving, it would be very important to ensure traffic laws are observed.
The proposed method does not seem to have a mechanism to ensure compliance with the law.
Is this a method that could be extended in the future?
I have questions about how these cars can comply with the law.

There appear to be several minor typographical and formatting errors in this paper.
The author could review the paper in detail and correct them. For example, the paper may not include a space before the citation number. A space between the citation number and the word preceding it should be included.

**Robotics Focus:**

Highly relevant to robotics but no hardware experiments

**Summary Of Paper:**

This paper proposed a method that combines Traffic Coordinator Network (TraCo), Counterfactual Advantage Function (CAF), and attention mechanism.
The proposed method is based on multi-agent reinforcement learning and aims to solve problems related to automated driving in multi-vehicle environments.
On busy roads, vehicles need to operate efficiently, which is also related to safety.
The experiment was compared to several baseline algorithms.
A simulator environment called METADRIVE was used.
Results showed the effectiveness of the proposed method.

**Summary Of Recommendation:**

There are some concerns with the lack of clarity in the claims made in the introduction of this paper and in the experiments.

---

### Decision · Program_Chairs · 2023-08-30

**Decision:**

Accept (Poster)

**Comment:**

This paper presentes a multi-agent RL method for autonomous driving in busy enviornment, featuring a traffic coordinator network.

The reviewers have agreed that the paper presents in interesting approach to a timely issue in the context of autonous driving. Most reviewers have found the approach being technically sound and well-described. However, there is a concern on the reality of the test scenarios in the paper, in the sense that the simulated environment does not capture the chain reaction of accidents in real driving conditions; this concern has not been completely resolved in the rebuttal phase.

While the AC thinks that the proposed scheme has limitation to be directly applied to real situations, he/she also thinks that the presented idea and the verification on a (limited) virtual environment would also be useful for the audience of CoRL. The authors need to incorporate the comments provided in their rebuttal in a final version.